# Dual Gene Detection of H5N1 Avian Influenza Virus Based on Dual RT-RPA

**DOI:** 10.3390/molecules29122801

**Published:** 2024-06-12

**Authors:** Qian Wang, Shiwen Wu, Jiangbing Shuai, Ye Li, Xianshu Fu, Mingzhou Zhang, Xiaoping Yu, Zihong Ye, Biao Ma

**Affiliations:** 1College of Life Sciences, China Jiliang University, Hangzhou 310018, China; s21090710055@cjlu.edu.cn (Q.W.); s22090710056@cjlu.edu.cn (S.W.); zmzcjlu@cjlu.edu.cn (M.Z.); yxp@cjlu.edu.cn (X.Y.); zhye@cjlu.edu.cn (Z.Y.); mb@cjlu.edu.cn (B.M.); 2Zhejiang Institute of Inspection and Quarantine Science and Technology, Hangzhou 311241, China; sjb@zaiq.org.cn; 3Zhejiang University of Science and Technology, Hangzhou 310023, China; liye20@zust.edu.cn

**Keywords:** dual RPA, avian influenza virus, H5N1 subtype, virus detection

## Abstract

The H5N1 avian influenza virus seriously affects the health of poultry and humans. Once infected, the mortality rate is very high. Therefore, accurate and timely detection of the H5N1 avian influenza virus is beneficial for controlling its spread. This article establishes a dual gene detection method based on dual RPA for simultaneously detecting the HA and M2 genes of H5N1 avian influenza virus, for the detection of H5N1 avian influenza virus. Design specific primers for the conserved regions of the HA and M2 genes. The sensitivity of the dual RT-RPA detection method for HA and M2 genes is 1 × 10^−7^ ng/μL. The optimal primer ratio is 1:1, the optimal reaction temperature is 40 °C, and the optimal reaction time is 20 min. Dual RT-RPA was used to detect 72 samples, and compared with RT-qPCR detection, the Kappa value was 1 (*p* value < 0.05), and the clinical sample detection sensitivity and specificity were both 100%. The dual RT-RPA method is used for the first time to simultaneously detect two genes of the H5N1 avian influenza virus. As an accurate and convenient diagnostic tool, it can be used to diagnose the H5N1 avian influenza virus.

## 1. Introduction

Avian influenza is a disease caused by infection with avian influenza virus (AIV), which belongs to the family Orthomyxoviridae [1]. According to 16 types of hemagglutinin (H1–H16) and 9 types of neuraminidases (N1–N9), avian influenza viruses are classified into different subtypes [2,3]. The genetic combinations of HA and NA subtypes of avian influenza viruses present in poultry are strictly limited, and the avian influenza viruses that cause poultry diseases are mainly H5 and H7 subtypes [4]. According to pathogenicity, avian influenza viruses can be divided into highly pathogenic and low pathogenic avian influenza viruses [5]. Most low pathogenic avian influenza viruses can cause mild respiratory, intestinal or reproductive diseases [6], while highly pathogenic avian influenza viruses are characterized by high incidence rate and mortality [7]. Among various highly pathogenic strains, the H5N1 avian influenza virus not only infects poultry but also humans, with a very high mortality rate [8], and is considered the most pathogenic avian influenza virus [9].

In 1996, the H5N1 avian influenza virus was first isolated from geese in Guangdong Province, China [10], and in 1997, it was first discovered in Hong Kong that the H5N1 avian influenza virus can be directly transmitted from poultry to humans [11]. After the first human infection with the H5N1 avian influenza virus, the virus gradually entered the public eye and spread globally [12]. In 2003, the outbreak in Southeast Asia quickly spread to Kazakhstan, Russia, Türkiye, Romania and other countries [13]. In addition, the H5N1 avian influenza virus has evolved into new highly pathogenic subtypes of avian influenza, such as H5N2, H5N5, H5N6, and H5N8, during its transmission process, and broke out in Asia, Europe, and North America in 2014, causing huge losses [14]. According to a report by the World Health Organization (WHO), there were a total of 868 cases of human infection with the H5N1 avian influenza virus from 2003 to 2023, of which 457 died, with a mortality rate of approximately 53% [15].

Currently, the degree of globalization is deepening, and the connections between countries are becoming increasingly close. Any outbreak of avian influenza in a country can easily affect other countries, resulting in serious economic losses. From 2013 to 2022, the import volume of poultry in China has grown rapidly (see Figure 1), with a sharp increase in testing volume. Therefore, high sensitivity, strong specificity, simple and accurate operation technology is needed in inspection and quarantine to control virus transmission, reduce economic losses and human infections [16]. At present, the nucleic acid amplification techniques for H5N1 avian influenza virus mainly include Reverse Transcription Polymerase Chain Reaction (RT-PCR) [17], Reverse Transcription Recombinase Polymerase Amplification (RT-RPA) [18], and Reverse Transcription Loop mediated Isothermal Amplification (RT-LAMP) [19]. RPA is a rapid isothermal nucleic acid amplification technique that utilizes recombinant enzymes obtained from bacteria or fungi [20]. Unlike PCR, RPA amplifies within a few minutes at a constant low temperature [21], and its technical core consists of recombinase, polymerase, and single stranded binding protein [22]. The amplification results of RPA can be directly observed by agarose gel electrophoresis [23], or combined with lateral flow chromatography strip [24]. The current detection methods mostly focus on a single target gene, which is not accurate enough for clinical diagnosis.

In this study, specific primers were designed for the HA gene and M2 gene of H5N1 avian influenza virus, and a dual gene detection method for H5N1 avian influenza virus was established based on dual RPA. Combined with agarose gel electrophoresis, the detection results can be observed under the chemiluminescence imaging device.

## 2. Results

### 2.1. Establishment and Optimization of Dual RT-RPA Detection

The amplification reaction was optimized with primer volumes of 3:2, 4:3, 1:1, 3:4 and 2:3 (HA:M2), respectively. The results showed that the amplification effect of HA gene was better in the ratio of 3:2 and 4:3. In the ratio of 3:4 and 2:3, M2 gene amplification effect is better; At the ratio of 1:1, the amplification effect of the two genes is comparable, so a 1:1 primer volume ratio is selected (Figure 2). The optimized reaction system is shown in Table 1.

The optimal reaction temperature of dual RT-RPA showed that the amplification could be performed from 37 °C to 42 °C, and the two bands were brightest at 40 °C, so 40 °C was the best reaction temperature (Figure 3).

The optimization results of the reaction time of double RT-RPA show that bands can be amplified after 10 min of double RT-RPA, and the bands become brighter and brighter with the increase of time. When the time exceeds 20 min, there is no significant difference in the brightness of the bands, so 20 min can be used as the optimal time (Figure 4).

### 2.2. Dual RT-RPA Sensitivity

In order to determine the detection limit of the detection method established in this paper, the standard plasmid was diluted 10-fold as the template, and the optimized reaction system and conditions were used for double RT-RPA detection. The amplified product was purified and then subjected to 3.5% agarose gel electrophoresis, and the results were observed under chemiluminescence imager. No amplification was detected at the plasmid concentration of 1 × 10^−8^ ng/μL, while amplification was detected at the plasmid concentration of 1 × 10^−5^ to 1 × 10^−7^ ng/μL. Therefore, the detection limit of dual RT-RPA was 1 × 10^−7^ ng/μL (Figure 5).

### 2.3. Dual RT-RPA Specificity

The established dual RT-RPA was used to detect H5N1, H1N1, H7N9, H3N2, H9N2 subtypes, Newcastle disease virus (NDV), and Infectious bruchitis virus (IBV), respectively. Target fragments were observed in samples of H5N1 avian influenza virus, but no amplification was seen in samples infected with other viruses (Figure 6). Therefore, the dual RPA detection method established in this paper has good specificity and no cross-reaction with other viruses.

### 2.4. Detection and Evaluation of Clinical Samples

To test the effectiveness of this method in clinical application, we extracted the RNA of H5N1 avian influenza virus, evaluated the performance of the dual RT-RPA method in 72 clinical samples, and compared it with RT-qPCR results. RT-qPCR was performed using PreScript One-Step RT-qPCR SYBR Green Kit (LMAI Bio, Shanghai, China). Among the 72 samples, RT-qPCR detected 28 positive samples (38.89%), and RT-RAA correctly identified 28 positive samples, with 100% sensitivity and 100% specificity (Table 2). At a 95% confidence level, the Kappa value of RT-RPA and RT-qPCR was 1 (*p* < 0.05). Based on the above data, plot ROC (Figure 7) to determine the detection accuracy of dual RT-RPA and qPCR. The results showed that AUROC was 1 (*p* < 0.001), indicating that our detection system method is an effective diagnostic tool that can accurately detect the H5N1 avian influenza virus.

## 3. Discussion

Avian influenza virus is distributed worldwide and has caused hundreds of thousands of zoonotic infections in recent years, with a high case fatality rate [25], posing a serious threat to human and animal health [26]. The H5N1 avian influenza that broke out in Hong Kong in 1997 [11], the H1N1 avian influenza that caused a pandemic in 2009 [27], and the H2N2 avian influenza that caused a serious impact in the United States in the middle of the last century [1] have all caused serious economic losses, but among them, the H5N1 avian influenza virus is the most serious and the most pathogenic [9]. Therefore, it is necessary to establish a rapid and accurate detection method for H5N1 avian influenza virus.

Accurate detection method is an effective means to prevent the spread of H5N1 avian influenza virus. In recent years, more and more detection methods have been applied to the detection of avian influenza virus, among which the isolation and identification of the virus is the gold standard for detection [28], but it needs to be carried out in the tertiary biosafety laboratory. Violetta Szczynska et al. [28] established an ELISA based avian influenza virus detection method, which can effectively detect avian influenza virus. Hua Bai et al. [29] established a surface plasma biosensor for detecting avian influenza virus. The emergence of these detection methods has contributed to better prevention of the spread of avian influenza virus.

RPA is a technology for constant temperature nucleic acid amplification using recombinase and polymerase. Recombinase can bind primer DNA to form protein-DNA complex, which can recognize homologous sequence and initiate chain replacement to form D ring. Single-chain binding protein can keep single chain stable and form new DNA chain under the action of DNA polymerase. The amplified products can grow exponentially [30]. Compared with PCR, RPA detection technology has the advantages of short reaction time, simple operation, high sensitivity and strong specificity [31]. Although it has only been developed for more than ten years, it has been applied to many fields such as SARS-CoV-2 [32], parasites [33] and Vibrio vulnificus [34]. However, RPA has been developed so far, there is still no special primer design software, and it can only be assisted by PCR primer design software. In addition, the products of RPA amplification must be purified before agarose gel electrophoresis, and compared with PCR, there is no thermal cycle step, and it is very easy to generate primer dimers.

## 4. Materials and Methods

### 4.1. Ethical Statement

Since all samples were virus samples archived in the repository of Zhejiang Provincial Institute of Inspection and Quarantine Technology, no samples were collected or animals were specifically slaughtered in this study. The parts of the tests involving virus samples were carried out in BSP Level 3 laboratories.

### 4.2. Extraction of Samples and Viral Nucleic Acids

Viral RNA was extracted using the RNAeasy™ viral RNA Extraction kit (centrifugal column) (Beijing Zhuangmeng Biotechnology, Beijing, China) according to the production manual. The extracted RNA was stored at −80 °C. cDNA was synthesized from viral RNA using PrimeScript™II Reverse Transcriptase enzyme kit (Nantong Feiyu Biotechnology Co., Ltd., Nantong, China) for subsequent detection.

### 4.3. Construction of Standard Plasmid

The reference strains Influenza A virus (A/goose/Guangdong/1/1996(H5N1)) (GenBank entry number: AF144305.1, AF144306.1) were used to prepare the standard products of HA gene and M2 gene of H5N1 avian influenza virus. Select PUC-SP carrier, Amp resistance, a 1707 bp fragment of H5N1 avian influenza virus HA gene was constructed, and a 982bp fragment of M2 gene was constructed. Synthesized by Shenggong Biotechnology (Shanghai, China) Co., Ltd. The two standard plasmids were mixed, and the standard plasmids were continuously diluted with RNaseH_2_O at a 10-fold gradient to a plasmid concentration of 1 × 10^−8^ ng/μL.

### 4.4. Primer Design

In order to detect H5N1 avian influenza virus, we obtained the HA gene sequence and M2 gene sequence of 100 H5N1 avian influenza viruses from GenBank database, used DNAMAN 6.0 software (LynnonBiosoft) for multiple sequence comparison, designed specific primers in the conserved regions of genes, and used NCBI BLAST function to further analyze the specificity of primers. qPCR and RPA primers were designed for HA gene and M2 gene respectively, and the best primers were selected, as shown in Table 3. All primers are synthesized by Shenggong BioEngineering (Shanghai, China) Co., Ltd.

### 4.5. Establishment of Dual RT-RPA Detection

This study used TwistAmp Basic Kit kit (Twistdx, Cambridge, UK) with a total volume of 50 μL. Including buffer and freeze-dried powder (mainly composed of recombinase, polymerase, single chain binding protein dNTPs), control template, mixed primers, MgOAc (280 mM). The reaction mixture contained the following: 2 μL template, 2.4 μL each primer (10 μM), 29.5 μL buffer, 2.5 μL MgOAc (280 mM), and 6.4 μL of DEPC water. The amplified product was purified with DNA product purification kit and then subjected to 3.5% agarose gel electrophoresis, 70 V electrophoresis for 100 min. After electrophoresis, observed the experimental results using a chemiluminescence imager. The brighter the band, the better the amplification effect. Then, used Gel Pro Analyzer (Media Cybernetics, Rockville, MD, USA) to analyze the results. Firstly, optimized the volume ratio of primers for the two genes. Further optimized the reaction temperature and amplification time.

### 4.6. Dual RT-RPA Sensitivity Analysis

To determine the detection limit of dual RT-RPA, a series of 10-fold dilutions were performed on positive plasmids containing HA gene and M2 gene fragments, with a dilution range of 1 × 10 ^−5^~1 × 10 ^−8^ ng/μL. Use DEPC water as the negative control.

### 4.7. Dual RT-RPA Specificity Analysis

In order to determine the specificity of double RT-RPA, RNA of avian influenza viruses H5N1, H1N1, H7N9, H3N2, H9N2 subtypes, Newcastle disease virus and avian infectious bronchitis virus in Table 4 were extracted as templates for double RT-RPA detection to verify the specificity of the method.

### 4.8. Evaluation of Dual RT-RPA Detection Methods

To conduct reliability testing on dual RT-RPA, a 95% confidence level was used for analysis to calculate the Kappa and *p*-values of RT-qPCR and dual RT-RPA. In addition, we also calculated the sensitivity and specificity of dual RT-RPA in clinical samples of H5N1 avian influenza virus. All statistical analyses were conducted using SPSS 26.0 (IBM) software.

## 5. Conclusions

In this study, dual gene detection of H5N1 avian influenza virus was established based on dual RT-RPA. The two sets of specific primers designed had good specificity. The optimal primers ratio of HA gene and M2 gene was 1:1, the optimal amplification temperature was 40 °C, and the optimal amplification time was 20 min. The sensitivity of dual RPA for detection of H5N1 avian influenza virus can reach 1 × 10^−7^ ng/μL. After the validation of clinical samples, at a 95% confidence level, the sensitivity and specificity are 100%, with good sensitivity and specificity. After Kappa consistency test, the coincidence rate with RT-qPCR detection is 100%. In summary, our dual RT-RPA can detect H5N1 avian influenza virus quickly and accurately, is expected to become a powerful and valuable detection tool.

## Figures and Tables

**Figure 1 molecules-29-02801-f001:**
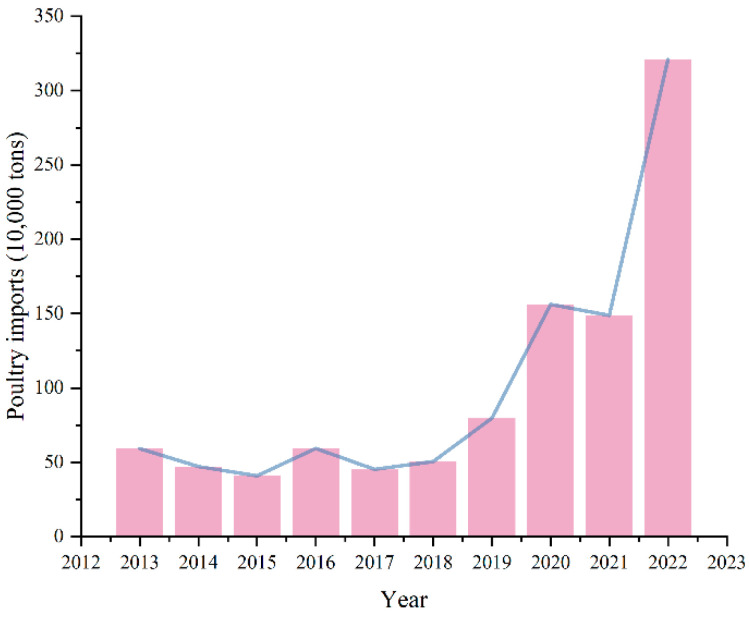
Import situation of poultry products in China from 2013 to 2022. (Data sourced from http://www.cvonet.com/data/list/4/8/33.html (accessed on 16 April 2024)).

**Figure 2 molecules-29-02801-f002:**
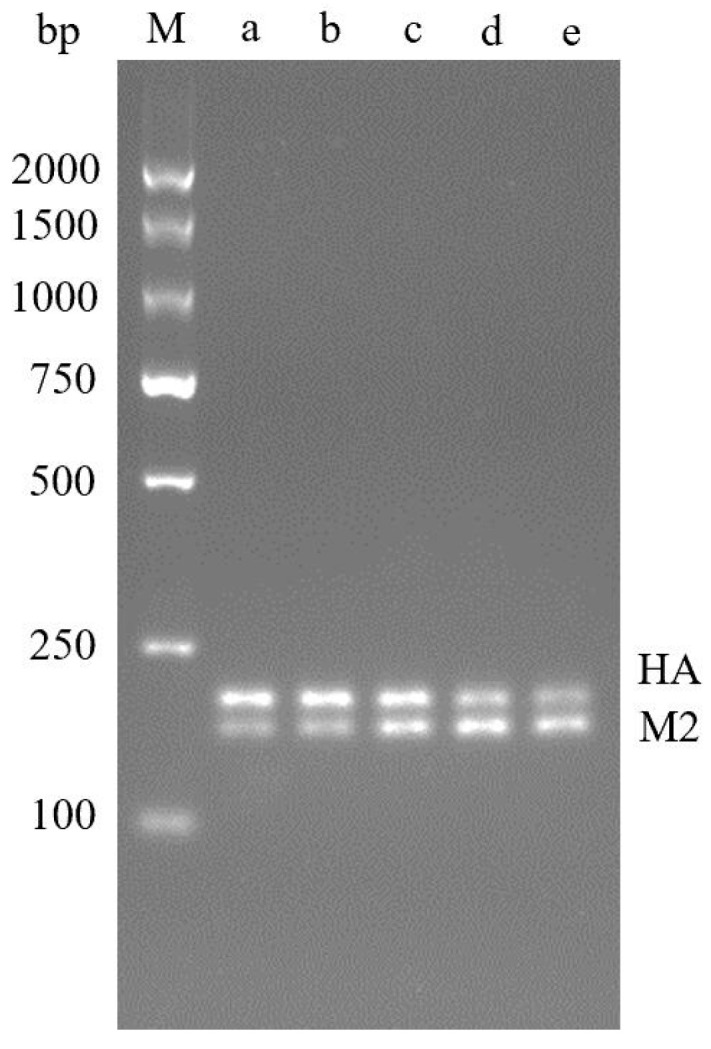
Optimization results of dual RPA primer volume ratio. **M**—DNA Marker DL2000; **a**—3:2; **b**—4:3; **c**—1:1; **d**—3:4; **e**—2:3.

**Figure 3 molecules-29-02801-f003:**
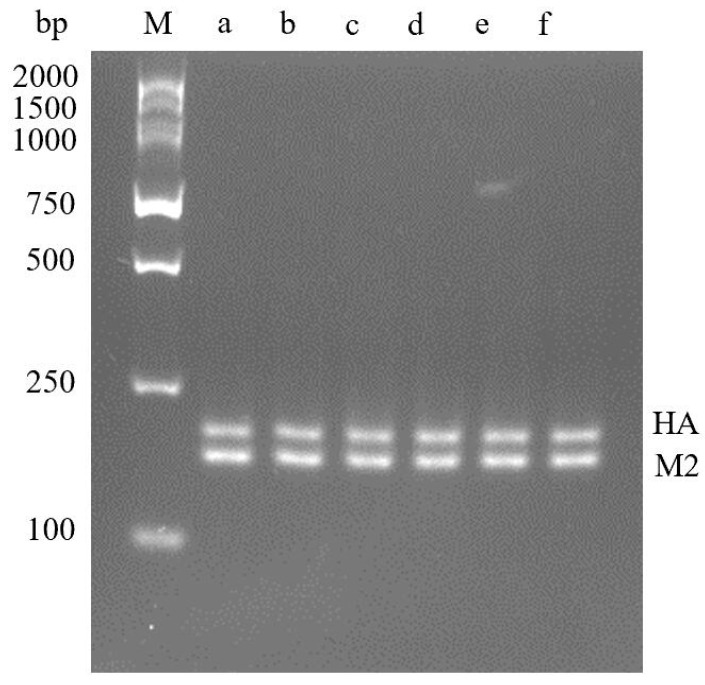
Optimization results of dual RPA reaction temperature. **M**—DNA Marker DL2000; **a**—37 °C; **b**—38 °C; **c**—39 °C; **d**—40 °C; **e**—41 °C; **f**—42 °C.

**Figure 4 molecules-29-02801-f004:**
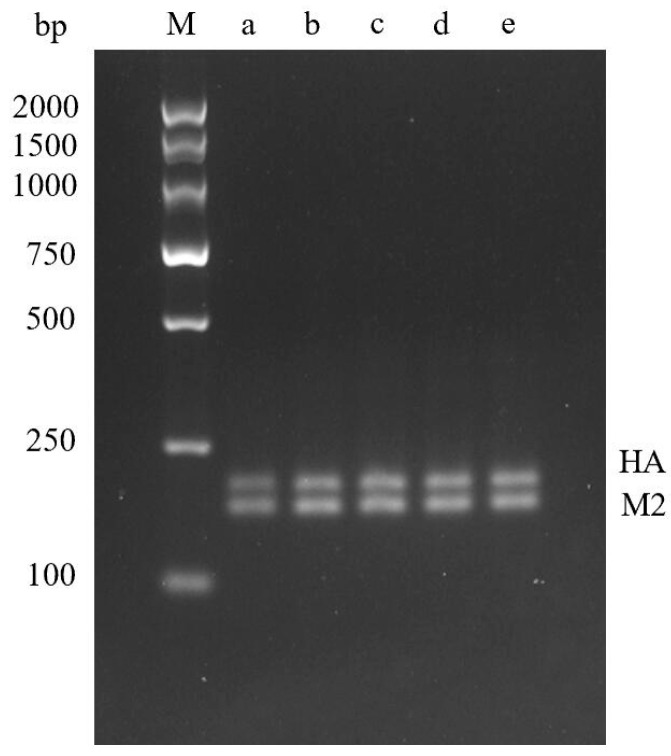
Optimization results of dual RPA reaction time. **M**—DNA Marker DL2000; **a**—10 min; **b**—15 min; **c**—20 min; **d**—25 min; **e**—30 min.

**Figure 5 molecules-29-02801-f005:**
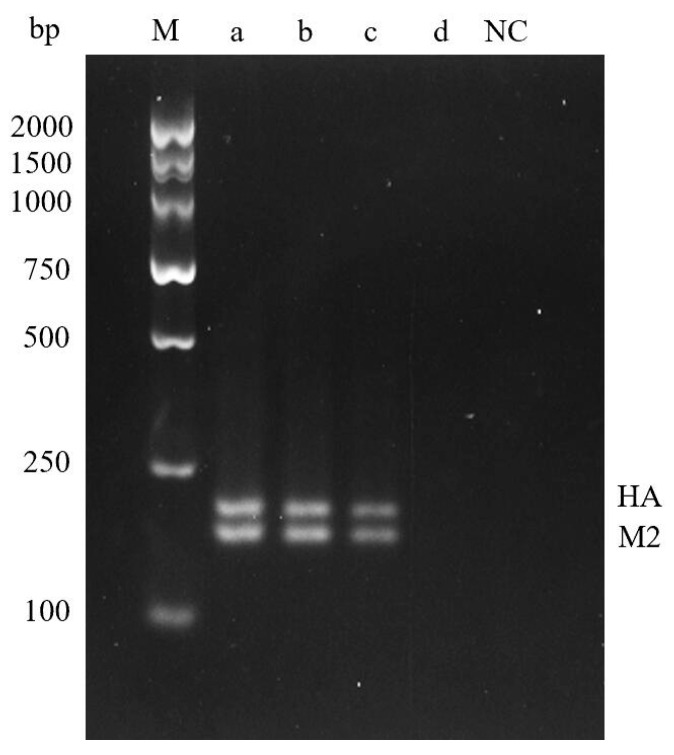
Dual RPA Sensitivity Detection Results. **M**—DNA Marker DL2000; **a**—1 × 10^−5^ ng/μL; **b**—1 × 10^−6^ ng/μL; **c**—HA: 1 × 10^−7^ ng/μL; **d**—1 × 10^−8^ ng/μL; **NC**—Negative control.

**Figure 6 molecules-29-02801-f006:**
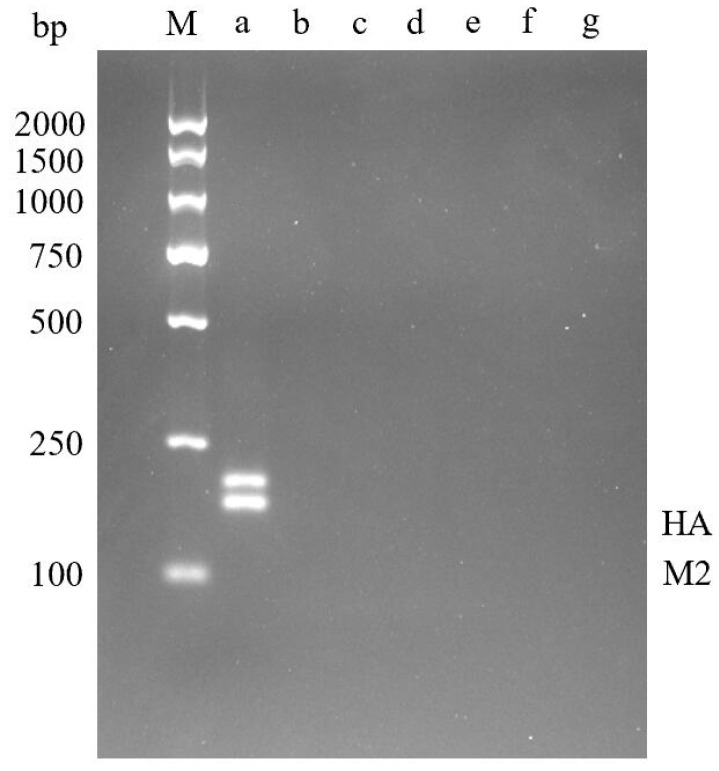
Dual RPA Specific Detection Results. **M**—DNA Marker DL2000; **a**—H5N1 AIV; **b**—H1N1 AIV; **c**—H7N9 AIV; **d**—H3N2 AIV; **e**—H9N2 AIV; **f**—Newcastle disease virus; **g**—Infectious bronchitis virus.

**Figure 7 molecules-29-02801-f007:**
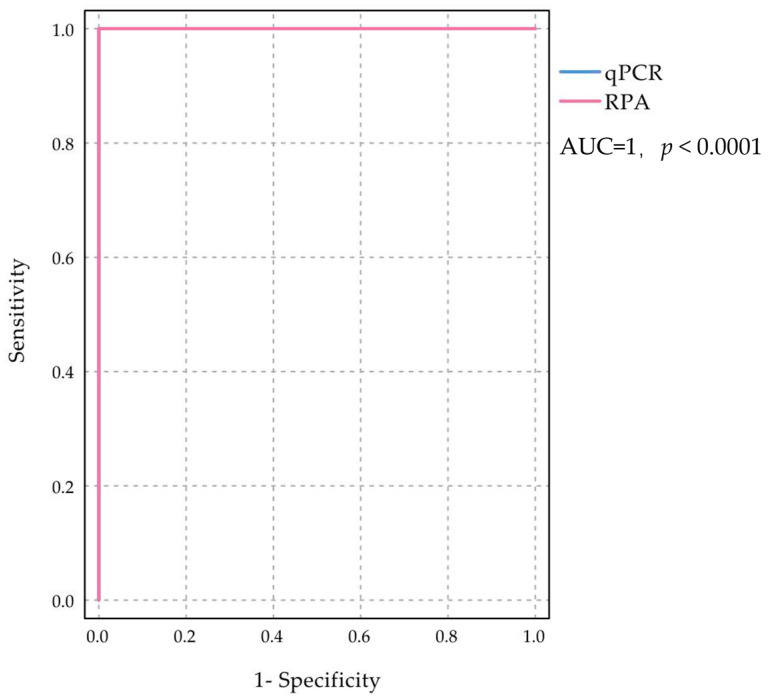
Diagnostic performance of the dual RPA.

**Table 1 molecules-29-02801-t001:** RPA amplification reaction optimized system.

Component	Volume (μL)
HA-RPA-F Primer (10 μM)	2.4
HA-RPA-R Primer (10 μM)	2.4
M2-RPA-F Primer (10 μM)	2.4
M2-RPA-R Primer (10 μM)	2.4
Buffer	29.5
template	2
DEPC H_2_O	6.4
280 mM MgOAc	2.5
Total	50

**Table 2 molecules-29-02801-t002:** Detection of H5N1 AIV in clinical samples.

		RT-RPA		Total	Kappa	Confidence Interval		*p* Value	Sensitivity %	Specificity %
		Positive	Negative			Upper Limit	Lower Limit			
RT-qPCR	Positive	28	0	28	1	0.041	0	<0.05	100	100
	Negative	0	44	44						
Total		28	44	72						

**Table 3 molecules-29-02801-t003:** qPCR and RPA primer sequences for HA and M2 genes.

Gene	Name	Primer Sequence (5′–3′)	Product Length (bp)
HA	HA-RPA-F	TCTAGTATGCCATTCCACAACATACACCCCC	193
HA	HA-RPA-R	AACCATCTACCATTCCCTGCCATCCTCCCTC	
M2	M2-RPA-F	AGCTATACAAGAAGCTGAAAAGAGAAATAAC	167
M2	M2-RPA-R	CTGCTCACAAGTGGCACACACTAGGCCAAAA	
HA	HA-qPCR-F	TCAAACTCCAATGGGGGCG	249
HA	HA-qPCR-R	CCCTGCTCATTGCTATGGTG	
M2	M2-qPCR-F	GGGATTTTAGGATTTGTG	350
M2	M2-qPCR-R	CTGATTAGTGGGTTGGTG	

**Table 4 molecules-29-02801-t004:** The name, subtype, pathogenicity, and other information of the virus used in this study.

**Sample**	**Virus**	**Subtype**	**Pathogenicity**	**Dual RT-RPA Assay**	**RT-qPCR**
K144(2.3.2.1)	AIV	H5N1	high	+	+
S1322(2.3.2)	AIV	H5N1	high	+	+
MK2	AIV	H5N1	high	+	+
BBVM	AIV	H5N1	high	+	+
390	AIV	H1N1	low	−	−
s1069	AIV	H7N9	high	−	−
X1330	AIV	H3N2	low	−	−
A32	AIV	H9N2	low	−	−
NX2006016	NDV	/	/	−	−
JS1816	NDV	/	/	−	−
M41	IBV	/	/	−	−
H52	IBV	/	/	−	−

## Data Availability

The data presented in this study are available on reasonable request from the corresponding author.

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
