# Peer review of "Dual Gene Detection of H5N1 Avian Influenza Virus Based on Dual RT-RPA"

_molecules, 2024, doi:10.3390/molecules29122801_

Round 1
Reviewer 1 Report
Comments and Suggestions for Authors
This manuscript describes only the analytical sensitivity and specificity of a dual RT-RPA isothermal assay for the H5N1 Influenza A virus (IAV). Isothermal amplification assay for H5N1 is not novel, however the authors combined two most used genes for IAV. The authors used a minimal sample size to evaluate the robustness of their assay. For analytical specificity of the assay, the authors failed to include some of the most common poultry pathogens such as Avian Paramyxoviruses -2, -3, and -6, Gallid herpesviruses, Mycoplasmas, Escherichia coli, Salmonella, Pasteurella, ornithobacteria, and Avibacterium paragallinarum. The discussion could have been better written, and the authors should have paid more attention to removing what appears to be a comment from the previous reviewer (Lines 137-140). The manuscript failed to provide a statistical analysis of the data presented. The developed assay must be evaluated for diagnostic sensitivity and specificity to routinely use this assay.
Minor comments:
·Provide a list of complete strain details used in this study, including the clade and genebank ID. How many of these strains were LPAI and HPAI?
·Line 112: "compared with …. higher." It is not fair to compare a test that was not performed in similar experimental conditions; please remove this sentence as this doesn't belong to the authors' results.
·Line 118: "….were detected by the established double RT-RPA." Re-write this sentence does not sound correct.
·Line 137-140: Check?
·Line 179 to 186: Did authors create plasmids for this study? If yes, please include detailed methodology as supplementary information.
·Line 190: Include software version used company details.
Comments on the Quality of English Language
can be improved
Author Response
Dear reviewer,
Thank you for your valuable feedback on the manuscript. We have made revisions to the manuscript based on your suggestions, and the main content is attached.
Warmly regards,
Associate Professor Fu

Reviewer 2 Report
Comments and Suggestions for Authors
The authors present an isothermic test that has some advantages, compared to other molecular tests. Therefore, it could be an interesting option to use for monitoring.
There are some key information that is not included in the manuscript.
Methodology is not complete.
I am confused. The ethical statement says that all samples were virus samples archived. But authors include in methods that they did viral RNA extraction from clinical sample.
Which kind of samples were used?
The sample size seems to be small for the conclusion. Review the precision of the estimates.
Review the redaction of the Establishment of dual RT-RPA detection. It looks like the instructions, but not a report.
TAble 4. Improve the title of the table. It has to be more specific.
Results:
Add confidence intervals to all estimates: kappa (include the actual p value). and sensitivity and specificity.
Based on what do you state that bright is the same? Did you use a software? If yes, which?
Discussion
Take out the first paragraph, which are indications.
Discuss which other alternatives are for primer designs and what is the role of the variants and mutations.
Include limitations of this study.
Abstract
include the actual p value.
The number of samples analyzed , also the type of sample.
Author Response

(The authors gave the same response as above.)

Round 2
Reviewer 1 Report
Comments and Suggestions for Authors
I agree authors used previously published manuscripts for specificity analysis; however, it's important to validate any diagnostic assay using some common field pathogens or samples with mixed infections to strengthen the assay design.
Line 118: "…. were detected by the established double RT-RPA."...I believe the authors assessed/analysed, but didn't detect NDV or IBV. Therfore please change accordingly.
Author Response
Dear Professor Reviewer,
Thank you for your valuable feedback. We have made revisions to the manuscript based on your feedback, and the specific content is attached.
Warmly regards,
Associate Professor Fu

Reviewer 2 Report
Comments and Suggestions for Authors
Regarding the methodology of dual RPA it still looks like they are instructions, for instance , "After electrophoresis, observe the experimental results using a chemiluminescence imager", it should be in past tense: After electrophoresis, we observed the results....
Regarding interval confidences requested, it also seems that they were not actually added, the response says “At a 95% confidence level, the Kappa value of RT-RPA ..." but I don´t see the interval.
Regarding my observation 7, they are not discussing the rol of mutations in primer designs. And in my last comment they haven´t added the actual p value.
Author Response

(The authors gave the same response as above.)
